# The Effect of the Microstructure and Viscosity of Modified Bitumen on the Strength of Asphalt Concrete

**DOI:** 10.3390/polym16101370

**Published:** 2024-05-11

**Authors:** Antonina Dyuryagina, Yuliya Byzova, Kirill Ostrovnoy, Alexandr Demyanenko, Vitaliy Tyukanko, Aida Lutsenko

**Affiliations:** Department of Chemistry and Chemical Technology, Manash Kozybayev North Kazakhstan University, Petropavlovsk 150000, Kazakhstan; adyuryagina@inbox.ru (A.D.); kostrovnoy@mail.ru (K.O.); demianenkoav@mail.ru (A.D.); vetal3333@mail.ru (V.T.); l-a.13@mail.ru (A.L.)

**Keywords:** microstructure of bitumen, viscosity, strength of asphalt concrete, mathematical modeling, surfactant, polymer, bitumen modification

## Abstract

The purpose of these studies was to establish the influence of the microstructural and rheological characteristics of modified bitumen compositions on the strength indicators of asphalt concrete. The effect of additives concentration on the rheological characteristics and microstructure of binary “bitumen–surfactant”, “bitumen-AG-4I”, and ternary “bitumen-AG-4I-AG-4I” systems has been studied. To assess the effect of bitumen dispersion on the physical and mechanical characteristics of modified asphalt concrete samples, the compressive strength value was determined. The following chemicals have been used as additives: the original product AS-1, industrial additive AMDOR-10, and used sealant AG-4I, a product based on polyisobutylene and petroleum oils. At an increased content of AG-4I (C ≥ 1.0 g/dm^3^) in ternary systems, the contribution of the emerging intermolecular polyisobutylene network to the development of structuring processes increases while the viscous effect of the surfactant AS-1 decreases. It has been established that the minimum size of bee-like bitumen structures (1.66 µm) is recorded with the joint presence of additives in the bitumen, AS-1 at a level of 1.0 g/dm^3^ and AG-4I at a level of 1.0 g/dm^3^. Under the same concentration regimes of the ternary bitumen composition, the maximum increase in compressive strength R*_D_* was achieved with the smallest size of bee-like structures of modified bitumen.

## 1. Introduction

Currently, the most common types of road surfaces are asphalt concrete pavements made from petroleum bitumen [1,2]. From the analysis of literature data, it follows that in the conditions of modern heavy and intense traffic, traditional asphalt concretes are very often unable to provide the required physical and mechanical properties of coatings and their durability, due to the unsatisfactory quality of the bitumen binder [1,3,4].

The problem of creating effective road surfaces is relevant from the standpoint of resource conservation, ensuring the reliability and durability of their operational characteristics. Due to the chemical nature and structural characteristics of asphalt concrete, the main way to increase the service life of coatings is to change the structure and properties of the organic binder by introducing various modifiers into the bitumen composition [5,6,7]. The modification of road bitumen, as evidenced by the experience of world practice [1], is characterized by the versatility and ramification of the methods of its technological implementation, the essence of which boils down to two main directions. One of the technological directions for improving the quality of bitumen is based on the introduction of surface-active modifiers [8,9]. Most often, cationic surfactants are used to ensure high adhesion between bitumen and the mineral component of the road surface [8,10]. Current trends in the development of asphalt concrete production technology are based on the improvement of these coatings through the widespread use of various polymer additives. This direction is based on the modification of bitumen with polymer materials, which give bitumen its characteristic physical and mechanical properties by creating a spatial structural network in the bitumen [1,5,6,7]. The cooperation of polymer and surfactant additives is an opportunity for a complex combination of their individual modifying effects, as well as a multiple increase in efficiency due to the synergistic effect. A limiting factor in the complex use of additives is a significant increase in the cost of modified bitumen binder. This problem can be solved by the use of effective surfactants synthesized from petrochemical waste and waste polymers (a solution of polyisobutylene in mineral oil) that require recycling as part of new materials [11,12]. This will make it possible not only to effectively modify asphalt concrete mixtures but also, at the same time, to reduce the huge volume of industrial waste and to reduce environmental problems of environmental pollution.

From a physicochemical point of view, modification of the binder with a polymer and surfactant is accompanied by a change in the heterophase, the dispersion of the bitumen system, and ends with the formation of a new, stable system under these conditions, differing from the original bitumen in spatial, rheological, structural, and mechanical characteristics of the modified compositions. The rheological assessment of the bulk properties of solutions provides an idea of the change in the intermolecular interactions of the system [13]. It has been established [14] that viscosity is very sensitive to the formation of self-organizing intermolecular assemblies of various types. The work of Yang and Pal proved that the interaction between anionic, nonionic, and zwitterionic surfactants and anionic sodium carboxymethylcellulose had a strong effect on the viscosity index of the system [15]. It has been revealed [16] that a change in the viscosity of aqueous solutions of surfactants and polymers is achieved through the interaction of polymers with surfactants, thus forming interconnected physical networks, where the main methods of interaction are electrostatic and hydrophobic forces. Lo et al. [17] studied solutions of hydrophobically modified 2-Hydroxyethyl Cellulose modified with a benzoyl moiety using viscometry and found a pronounced viscosity maximum with the addition of SDS (sodium dodecyl sulfate). Rheological studies of polymer solutions have shown that the addition of SDS or DTAB (dodecyltrimethylammonium bromide) surfactants results in a 10- to 1000-fold increase in viscosity, while even higher concentrations of surfactant again reduce the viscosity index [18]. Despite many different works in this direction, they are mainly concerned with aqueous solutions of surfactants and polymers.

The possibility of a more detailed study of the microstructure of bitumen compositions is provided by atomic force microscopy (AFM), which allows to obtain statistically reliable information about the morphological features of the emerging structures [19,20], ultimately influencing the performance properties of the bitumen binder. Previously, the authors investigated the effect of the microstructure of the interfacial zone between filler particles and bitumen binder on the strength of the asphalt concrete mixture and found that the modulus and hardness of asphalt mortar decrease with the increases in the aggregate surface distance and then keep stable [21]. As is known, not only does the microstructure of the interfacial layers between the solid particles of the filler and bitumen affect the performance characteristics of asphalt concrete, but also the size of the aggregates in the volume of the dispersed structure of the binder. Leber, who discovered in 1996 a structure similar to the body structure of a bee in an AFM image of bitumen, [22] called it a “bee-like structure”. Currently, there is still no consensus on the mechanism of formation of bee-like structures. The relationship between the microstructural characteristics of bitumen and the structural and mechanical properties of bitumen has become a relevant direction in the study of the performance characteristics of asphalt concrete.

Zhang et al. [23] analyzed the correlation between bitumen microstructure parameters and the main characteristics of SBS-modified bitumen pavement. It was found that penetration decreases and softening temperature increases with increasing area fraction of the bee structure of SBS-modified bitumen (Styrene Butadiene Styrene). In another work, Zhang et al. found that during the aging process of bitumen, with an increase in microstructural parameters, there is a decrease in penetration at 25 °C and ductility of bitumen at 15 °C and an increase in the softening temperature and viscosity of the binder at 60 °C [24]. In this regard, the study of the patterns of structuring the bulk phase of a dispersed system in the presence of polymer and surfactants will allow directional control of the dispersion and performance properties of bitumen materials. However, the use of additives is complicated by the fact that the generation of specified surface and bulk properties of bitumen is carried out not chemically but through transformations realized at the level of intermolecular interactions. The narrow range of energy values within which various states of easily transformed modified forms are realized requires targeted selection and optimal dosage of additives to achieve the necessary performance properties of the formed road surfaces, including maximum strength of asphalt concrete.

Thus, the purpose of this research was to establish the influence of the microstructural and rheological characteristics of modified bitumen compositions on the strength indicators of asphalt concrete. This required studying the changes in viscosity and the nature of the distribution of microdispersions in binary and triple bitumen compositions from the concentration of additives, as well as determining the strength characteristics of modified asphalt concrete.

## 2. Materials and Methods

### 2.1. Materials

For conducting research, the following materials were used:Oxidized bitumen with penetration 100/130 (bitumen brand BND 100/130—oil bitumen for roads in Kazakhstan), manufactured by Gazpromneft-Bitumen Kazakhstan LLP, Shymkent, Kazakhstan.Modifying additives are as follows:
-AS-1 [25] is a product of the amination of distillation residues of petrochemistry (KON-92), which is a mixture of amines of the general formula
R′-NH_2_, R′-NH-R″,
where R′ is n-butyl, R″—2-ethyl-2-hexenyl;-AG-4I is a used sealant, a product based on high molecular weight polyisobutylene (PIB) and petroleum oils (manufactured by “Germetika Research and Production Company”, Moscow, Russia);-AMDOR-10—a mixture of polyaminoamides and polyaminoimidazolines (manufacturer CJSC “Amdor”, Saint-Petersburg, Russia), a condensation product of polyamines and higher fatty acids.



### 2.2. Methods

The method of preparation of modified bitumen compositions was described in detail earlier in the article [26].

#### The Method of Measuring the Viscosity of Modified Bitumen Compositions

The absolute viscosity of modified bitumen compositions (t = 130 °C) was measured using a viscometer of the VZ-DIN-4 brand (“AnalytPromPribor’’ LLC, Moscow, Russia). The flow time of bitumen with a volume of 50 mL through a hole with a diameter of 4.0 mm was taken as an indicator of absolute viscosity. A brass insert with a fixed diameter of the drain hole was inserted into the viscometer and closed with a valve. Bitumen heated to the required temperature was poured into the cylinder of a viscometer, the liner was pushed aside, and the stopwatch was turned on.

The viscometer constant (*k*) was determined by the flow time of distilled water at a fixed temperature of 25 °C.
(1)$$\eta_{comp}=k\frac{\tau_{comp}}{\tau_{H_{2}O}},$$
where *η_comp_*—absolute viscosity of the composition;
*τ_comp_*—the flow time of the composition;*τ*_*H*_2_*O*_—the flow time of distilled water.

The values of the viscosity were calculated based on the results of averaging the results of measuring the flow time of solutions obtained in three series.

In addition, for a comparative assessment of the rheological properties of modifiers in binary and ternary systems, the relative (2) and specific viscosity (3) were calculated.
(2)$$\eta_{rel}=\frac{\eta_{comp}}{\eta_{bitumen}}$$
where *η*_rel_—relative viscosity,
*η_bitumen_*—absolute viscosity bitumen.
(3)$$\eta_{sp}=\eta_{rel}-1$$
*η_sp_*—specific viscosity of the composition.


### 2.3. The Method of Analyzing the Microstructure of Modified Bitumen Compositions by Atomic Force Microscopy

The AFM method makes it possible to obtain a digital model of the topographic surface of bitumen in the form of an image to establish the effect of the studied surfactant (AS-1) and polymer (AG-4I) additives on the microstructure of the bitumen.

The study of the microstructural organization of modified bitumen samples was carried out on a universal scanning device, Solver Spectrum (“NT-MDT” LLC, Moscow, Russia), in semi-contact mode using NSG-01 probes (“Tipsnano” LLC, Tallinn, Estonia). A bitumen sample was applied to a glass substrate and heated to spread and obtain a smooth surface. The temperature during the measurements was maintained within 20 ± 2 °C.

The maximum scanning area (Figure 1) was 50 × 50 µm (with a resolution of 300 × 300 pixels). Seven compositions of modified bitumen and a control composition without additives were prepared for the study (Table 1).

### 2.4. Method for Determining the Compressive Strength of Asphalt Concrete Samples

To assess the effect of bitumen dispersion on the physical and mechanical characteristics of modified asphalt concrete samples, the IP-100M-auto test press (“Ctimul” LLC, Moscow, Russia) was applied to determine the compressive strength (R*_D_*). The compressive strength indicates the maximum pressure that an asphalt concrete material can withstand under vertical compression. This indicator is determined at 20 °C and 50 °C according to ST RK 1218-2003, “Materials based on organic binders for road and airfield construction”.

### 2.5. Modeling Method

Modeling of the effect of additive concentrations on the absolute viscosity of the triple system “bitumen-AG-4I-AS-1” at a temperature of 130 °C was carried out using the probabilistic deterministic planning method [27]. This method includes the design of an experimental plan based on a Latin or Greek-Latin square, conducting an experiment by entering the experimental data obtained into the experimental plan table, sampling the experimental array on dot graphs to identify partial dependencies, approximating the partial dependencies obtained in tabular and graphical form using mathematical equations and developing a mathematical model based on the Protodyakonov generalized equation using mathematically described partial dependencies. The adequacy of the obtained mathematical model could be confirmed by the coefficient of nonlinear multiple correlation, the significance of which is determined by the Fisher or Student criterion. A detailed description of the method and examples of its application are presented in the articles [26,27,28,29].

## 3. Results and Discussion

### 3.1. Rheological Characteristics of Binary Bitumen–Additive Systems

The results of studying the rheological properties of bitumen compositions at 130 °C depending on the concentration of modifiers (0–2.0 g/dm^3^) are presented in Table 2.

As evidenced by experimental data obtained at t = 130 °C, the introduction of 0.5 g/dm^3^ of all types of additives into the binder leads to a decrease in viscosity values relative to unmodified bitumen (Table 2).

The greatest decrease in specific viscosity in isoconcentration (C = 0.5 g/dm^3^) binary compositions occurs in the presence of AS-1 (*η*_sp_ = −0.62), i.e., the lowest molecular weight additive (M = 250 amu). As the molecular weights of the modifiers increased, the depth of viscosity changes decreased. Thus, in the composition with AMDOR-10 (M = 2260 amu), the specific viscosity had a value of −0.48, and with the higher molecular weight of AG-4I (M = 5400 amu), the decrease of relative viscosity (*η*_sp_ = −0.28) was even less (Figure 2).

With further dosing of additives into bitumen (C_M_ ≥ 1.0 g/dm^3^), the concentration dependence of the specific viscosity of the composition with a sealing liquid (Figure 2, curve 1) and with amphiphilic compounds (Figure 2, curves 2 and 3) have a completely different character and reflect two different mechanisms of intermolecular and interfacial interactions in the volumetric phase of bitumen. So, if with AG-4I (Figure 2, curve 1) the increment in bitumen viscosity due to the dissolved substance in the concentration range from 1.0 to 2.0 g/dm^3^ increases positively and invariably, then with AS-1 and AMDOR-10 it is the same and does not go into the region of positive values (Figure 2, curves 2 and 3). These differences are associated both with the dispersed structure of bitumen (core-asphaltenes, shell-resins, medium-oil) [1] and with the characteristics (composition, structure, spatial orientation) of the modifiers themselves.

Based on the results of numerous studies [1,30,31,32,33], it is known that the structural unit of resinous-asphaltene substances is condensed benzene rings with heteroatoms, forming a flat geometric figure with side substituents in the form of alkyl chains and naphthenic rings. Located parallel to each other, such structures form microassociates (packs, globules, micromicelles), separated from each other by an oil layer. When AG-4I (a solution of polyisobutylene in petroleum oil) is introduced into bitumen, the oil layer expands, which causes a restructuring of microassociates, and some of the molecules pass into the solution, providing the entire system with surface activity, which we previously proved by measuring surface tension [26]. As a consequence, the destruction processes developed under the influence of petroleum oils cause the observed decrease in specific viscosity in this concentration area (*η*_sp_ = −0.28). In parallel with this process, another component of AG-4I—polyisobutylene, spreading between bitumen microassociates, forms an intermolecular polymer network inside the bulk phase of the dispersed system. The development of homoassociation processes in the concentration range of 1.0–2.0 g/dm^3^ AG-4I (Figure 2, curve 1) is confirmed by the results of viscometric analysis: specific viscosity increases from 0.02 to 0.74, i.e., almost 37 times, which corresponds to the results obtained in works [14,34].

With the introduction of one of two varieties of amine-containing modifiers (AS-1 or AMDOR-10), the restructuring of the colloidal system of bitumen occurs according to a different mechanism. Part of their molecules concentrate at the bitumen–air interface, reducing surface tension [26], and the rest are sorbed on bitumen microassociates. Heteroassociation processes weaken the energy of intermolecular interactions in bitumen microassociates, which causes their destruction. This is evidenced by the obtained viscosity indicators of binary compositions with both types of amino derivatives, both AS-1 and AMDOR-10 (Table 2), which are invariably lower than the viscosity indicator of unmodified bitumen throughout the entire range of concentrations studied. The observed fluctuations in specific viscosity are associated with the reversibility of intermolecular formations, the equilibrium characteristics of which are determined by the quantitative content of amphiphilic compounds in the volume of the dispersed system. The lowest degree of destruction was noted in compositions with a modifier concentration of 1.0 g/dm^3^, which corresponds to the maximum of their surface activity [26]. This content corresponds to the highest specific viscosity values. The values of *η*_sp_ for AS-1 and AMDOR-10 were −0.31 and −0.36, respectively (Figure 2, curves 2 and 3). The maximum effect of destructuring (*η_s_*_p_ = −0.62) in compositions of the low-molecular AS-1 manifested itself at C_M_ = 0.5 g/dm^3^. In compositions of the high-molecular AMDOR-10, the maximum of this effect (*η_sp_* = −0.54) was observed at a concentration three times higher (C_M_ = 1.5 g/dm^3^). The latter is apparently associated with the conformational features of AMDOR-10 macromolecules, which cause steric hindrance for all nitrogen atoms to take part in adsorption processes (Figure 2, curve 3).

Judging by the concentration dependence of the specific viscosity of the binary composition “bitumen-AG-4I” (Table 3), in the range of additive concentrations C_M_ ≤ 0.5 g/dm^3^, the increasing discreteness of the bitumen system stimulated a decrease in destructuring processes; *η_sp_* has a value of −0.26 (Figure 2). A similar pattern in the area of limited concentrations of additives (C_M_ ≤ 0.5 g/dm^3^) was previously stated in the compositions “bitumen-AS-1” and “bitumen-AMDOR-10” (Table 4). However, in contrast to these compositions with amphiphilic additives, in the “bitumen-AG-4I” compositions, with a further concentration of the sealing liquid (C_M_ > 0.5 g/dm^3^), the specific viscosity invariably increased up to the final (C_M_ = 2.0 g/dm^3^) content (Table 3).

Another important distinctive feature of the development of the process of homoassociation of polyisobutylene macromolecules was that each subsequent dosing of an additional 0.5 g/dm^3^ of AG-4I into bitumen was accompanied by a gradual increase in specific (Δ*η_sp_* = 0.28–0.38) (Table 4).

### 3.2. Rheological Characteristics of Ternary Systems “Bitumen-AG-4I-AS-1”

The results of rheological studies in mixed compositions, including the joint presence of two modifiers in bitumen (AG-4I and AS-1), are presented in Figure 3. The effect of surfactant concentration on viscosity was determined at a fixed content of AG-4I in bitumen (C_M_ = 0.5–2.0 g/dm^3^) at t = 130 °C.

As a quantitative measure of the processes developing in the volume, along with the experimental indicators (*η_sp_._exp_*), their calculated values for the ternary system (*η_sp.calc_*) were used as a total value, taking into account the separate contribution of AG-4I and AS-1, i.e., in the absence of intermolecular interactions between them:(4)$${ƞ}_{sp.calc}={ƞ}_{sp.AG-4I}+{ƞ}_{sp.AS-1}$$
where *η*_sp.AG-4I_—specific viscosity at a fixed content of sealing liquid in the “bitumen-AG-4I” system;
*η*_sp.AS-1_—specific viscosity with variations in AS-1 concentrations in the bitumen-AS-1 system.


The data obtained show that although the experimental concentration dependences of the specific viscosity of AS-1 in the presence of AG-4I (Figure 3a–d, curves 1) are largely similar in appearance to the calculated ones (Figure 3a–d, curves 2), they are, however, shifted towards more high values of *η_sp_*.

Significant deviations from the additivity rule indicate an increase in the cooperative association of surfactant molecules with bitumen microassociates, which causes an increase in viscosity in the “bitumen-AG-4I-AS-1” systems. The strengthening of the association is associated with the presence in these systems, in addition to the introduced surfactant (AS-1), bitumen surfactants delocalized under the influence of the sealing liquid [26]. Part of them are adsorbed on the bitumen–air surface, reducing surface tension, and the rest are concentrated on bitumen microassociates, increasing intermolecular interactions and, accordingly, *η_sp_*. As experimental data show, for the formation of cross-links due to the cationic substance AS-1 and predominantly anionic bitumen surfactants, i.e., to manifest the viscous effect, a certain concentration ratio of the components is required. Depending on this, the association may increase or decrease.

At limited contents of AG-4I (C_M_ < 1.0 g/dm^3^), surfactants have a priority effect on association processes, and at increased concentrations (C_M_ ≥ 1.0 g/dm^3^), the structuring role of the formed intermolecular polyisobutylene network increases.

The greatest influence of surfactants on the viscosity effect was noted in the composition “bitumen-AG-4I-AS-1” with a minimum content of AG-4I (0.5 g/dm^3^). In this system (Figure 3a, curve 1), the extremum in specific viscosity (*η_sp_* = 0.34), manifested at C_AS-1_ = 1.0 g/dm^3^, significantly exceeds the value of the extremum at a similar concentration of AS-1 (*η_sp_* = −0.31) in the binary system “bitumen-AS-1”. This is clear evidence of the additional impact on the association processes of bitumen surfactants released by the sealing liquid. Further introduction of AS-1 (C > 1.0 g/dm^3^) is excessive and has a destructuring effect, even at C_M_ = 2.0 g/dm^3^, specific viscosity values in triple (*η_sp_* = −0.41) and binary (*η_sp_* = −0.43) systems are practically leveled off (Table 3 and Table 5).

With an increase in AG-4I consumption by two times (1.0 g/dm^3^), the maximum in specific viscosity values (*η_sp_* = +0.15) was observed at a higher concentration of AS-1 (2.0 g/dm^3^), and it was lower by 0.16 compared to the previous system (Figure 3b, curve 1). The decrease in the viscosity effect of the surfactant is associated with the competing influence of the intermolecular network of polyisobutylene, the formation of which was previously recorded in the concentration range of 1.0 ÷ 2.0 g/dm^3^ AG-4I in bitumen.

This becomes more obvious in the composition “bitumen-AG-4I-AS-1” containing 1.5 g/dm^3^ AG-4I (Figure 3c, curve 1). In this system, the introduction of AS-1 has practically no effect on the viscosity effect in the concentration range from 0.5 to 1.5 g/dm^3^. Each additional dosage of 0.5 g/dm^3^ AS-1 led to only minor (Δ*η_sp_* = ±0.07) deviations of the specific viscosity of the composition from the initial value (Δ*ƞ_sp_* = +0.36), determined by the content (1.5 g/dm^3^) AG-4I (Table 4). Only in a narrow range of elevated concentrations of AS-1 (from 1.5 to 2.0 g/dm^3^) is the viscous effect of the surfactant manifested, the intensity of which (Δ*ƞ_sp_* = 0.13) is minimal.

The viscosity effect of the surfactant is completely neutralized in a system containing 2.0 g/dm^3^ AG-4I (Figure 3d, curve 1). In a composition with a maximum content of free bitumen surfactants, when they, as was established earlier [26], are absent at the interface between bitumen and air and are completely concentrated in the volume of the disperse system, the additional introduction of 0.5 g/dm^3^ AS-1 causes a significant decrease in specific viscosity (Δ*ƞ_sp_* = −0.28) relative to the initial value (*η_sp_* = +0.74). However, this change in specific viscosity is still less than calculated (Figure 3d, curve 2), as is the intensity of its subsequent concentration fluctuations.

### 3.3. Modeling of the Effect of Additives Concentration on the Absolute Viscosity of a Bitumen Mixture in Triple System “Bitumen-AG-4I-AS-1”

To determine the effect of the concentration of additives at a temperature of 130 °C on the viscosity (*η*, s) of triple systems “bitumen–AG-4I–AS-1”, experiments were carried out according to the orthogonal 5 × 5 plan presented in Table 6. The input parameters of experimental studies are as follows:(1)Concentration of polymer AG-4I (C_AG-4I_, g/dm^3^), taking values 0, 0.5, 1.0, 1.5, or 2.0;(2)The concentration of surfactant AS-1 (C_AS-1_, g/dm^3^), taking the values 0, 0.5, 1.0, 1.5, or 2.0.

Table 6 shows the orthogonal plan of the experiment and the results of viscosity measurements. This plan considers all possible combinations of concentrations of additives AG-4I and AS-1 at a temperature of a bitumen mixture of 130 °C.

In accordance with the method of probabilistic deterministic planning [27], a sample of the experimental array (Table 5) on partial dependencies in tabular form (Table 7 and Table 8) was carried out, and the graphs of partial dependencies were obtained (Figure 4).

Analyzing the behavior of additives when they are used together in a triple system “bitumen-AG-4I-AS-1” at 130 °C (Figure 4), it is possible to note a decrease in the viscosity of the bitumen mixture with an increase in the concentration of any of the additives in the range from 0 to 0.5 g/dm^3^. It can also be noted that the concentration of AS-1 above 0.5 g/dm^3^ has an insignificant effect on the viscosity value. At the same time, with an increase in the concentration of AG-4I from 0.5 to 2.0 g/dm^3^ (Figure 4), the viscosity increases significantly (almost 2.5 times).

The partial dependencies shown in Figure 4 were approximated by second-order polynomials. Using the obtained polynomials, a two-factor mathematical model based on the generalized Protodyakonov equation [27] was developed. This model describes the combined effect of the concentration of the polymer AG-4I and the concentration of surfactant AS-1 on the viscosity of the bitumen mixture at a temperature of 130 °C (5):(5)$$\eta =\frac{\left(7.5429\cdot C_{AG-4I}^{2}+16.274\cdot C_{AG-4I}^{\;}+35.691\right)\cdot \left(7.2571\cdot C_{AS-1}^{2}-17.114\cdot C_{AS-1}^{\;}+69.509\right)}{63.28}$$

The coefficient of nonlinear multiple correlation [27] for this two-factor mathematical model is 0.94. The significance of the obtained nonlinear multiple correlation coefficient was confirmed using the Fisher criterion [27].

According to the obtained two-parameter mathematical model, nomograms were constructed for the fixed temperature of a bitumen mixture equal to 130 °C (Figure 5). These nomograms allowing to achieve the required viscosity value (*η* takes values of 40, 50, 60, 70, or 80 s, respectively) at different ratios of polymer (C_AG-4I_, g/dm^3^) and surfactant (C_AS-1_, g/dm^3^) concentrations. Nomograms allow to select the optimal concentrations of additives to achieve the required viscosity of the bitumen mixture.

Using the obtained two-factor mathematical model, nomograms were constructed to determine the optimal concentrations of polymer AG-4I and surfactant AS-1 to achieve a given viscosity value of the bitumen mixture (Figure 5). Analyzing the obtained nomograms (Figure 5), it can be noted that at a fixed concentration of the polymer AG-4I, the given viscosity value, in general, can be achieved at two concentrations of surfactant AS-1. If one value of the concentration of surfactant AS-1 less than 1.0 g/dm^3^ for a certain fixed value (less than 0.6 g/dm^3^) was selected, then the second value of the concentration of surfactant AS-1, corresponding to a given value of viscosity and concentration of polymer AG-4I, will be higher than the concentration of 1.4 g/dm^3^ by approximately the same certain fixed value. For example, the concentration of surfactant AS-1 equal to 0.8 g/dm^3^ will approximately correspond to the concentration of surfactant AS-1 equal to 1.6 g/dm^3^, and the concentration of 0.5 g/dm^3^ will approximately correspond to the concentration of surfactant AS-1 equal to 1.9 g/dm^3^. The maximum concentration of AG-4I at a given viscosity value corresponds to the concentration of AS-1 in the range of approximately 1.1 to 1.3 g/dm^3^.

### 3.4. Study of the Microstructure of Modified Bitumen

The results of AFM of binary bitumen compositions are presented in Figure 6 as well as diagrams of the distribution of bitumen microdispersions in the original bitumen and modified samples (Figure 7).

Comparing images Figure 6a–c it can be noted that when the polymer additive AG-4I is introduced into bitumen, a significant change in the microstructure of the bitumen occurs (Figure 6). As follows from the distribution diagram of bitumen microdispersions, in the absence of an additive (Figure 7, composition No 1), fractions ranging in size from 2.0 to 4.0 microns (P = 43.40%) predominate. The content of fine fractions (particle size ≤ 2.0 µm) does not exceed 24.50%, and large aggregates, larger than 6.0 µm, account for 17.10%. The average size (*a*_av_) of bee-like structures in unmodified bitumen was 3.69 µm.

When 0.5 g/dm^3^ AG-4I was introduced into bitumen, the average particle size decreased to 2.23 µm, and the content of fine fractions (≤2.0 µm) increased 1.7 times and amounted to 42.20% (Figure 7, composition No 2). The presented diagram clearly demonstrates that AG-4I increases the number of small (≤2.0 µm) and medium (≤2.0–4.0 µm) particles due to the absolute destruction of the largest dispersions (>6.0 µm) and partially aggregates ranging in size from 4.0 to 6.0 µm. In compositions that are more concentrated (C_AG-4I_ = 2.0 g/dm^3^) in terms of AG-4I content, the effect of the destruction of large bitumen microdispersions (>6.0 µm) remains approximately at the same level: the content of small fractions (≤2.0 µm) increased by 1.5 times and reached 37.20%, and the average particle size was 2.38 µm (Figure 7, composition No 3). The data obtained serve as a direct confirmation of the above results of rheological studies in isoconcentration binary compositions “bitumen-AG-4I” (Figure 2).

The introduction of an amphiphilic modifier (AS-1) into bitumen causes a different change in the dispersed composition (Figure 8).

As a result of the adsorption-propagating action of AS-1 at the same additive concentration (C_AS-1_ = 0.5 g/dm^3^), the size of the dispersions decreased only to 3.14 µm, and the content of fine fractions (≤2.0 µm) increased to 32.2% (Figure 9, composition No 2).

With an increase in the concentration of AS-1 from 0.5 to 1.0 g/dm^3^, large fractions (>6.0 µm) were destroyed mainly to aggregates measuring 2.0-4.0 µm (ΔP = 11.60%) with a very slight change (ΔP = 2.20%) in fine fractions (compared to the base version without surfactants). The average size of the dispersions was 3.14 µm. A comparison of the presented diagrams (Figure 7 and Figure 9) shows that, unlike AG-4I, under the influence of AC-1, there is no complete destruction of large fractions (>6.0 µm). The observed change in the dispersed composition in the binary compositions “bitumen-AS-1” is consistent with the results of viscometrical analysis (Figure 2, curve 2).

Visualization of the restructuring of the dispersed system of bitumen in the mixed composition “bitumen-AG-4I-AS-1” is presented in Figure 10.

Based on the results of AFM analysis, it was established that the intensity of the processes of dispersion of microdispersions in the joint presence of both modifiers is maximum at the concentration of C_AG-4I_ = 1.0 g/dm^3^ and C_AS-1_ = 1.0 g/dm^3^ (Figure 11); the average size of the dispersions was 1.66 µm, which is the minimum value in the entire studied range of modifier contents.

As a result of simultaneous exposure to AG-4I and AS-1 in these concentration regimes (Figure 11, composition No 4), not only fractions with sizes from 4.0 to 6.0 µm and higher (>6.0 µm) were completely destroyed, but more than two times much smaller aggregates (2.0–4.0 µm). As a result, in comparison with the base option, the content of fine fractions increased by 57.4% (P = 24.5–81.9%).

To assess the effect of bitumen dispersion on the physical and mechanical characteristics of asphalt concrete pavements, modified bitumen–mineral compositions were studied for compressive strength (Table 9).

From the data obtained, it follows that asphalt mixture samples using binary “bitumen–additive” compositions have improved strength characteristics, as evidenced by an increase in the values of compressive strength at 20 °C and 50 °C. Asphalt concrete samples made on the basis of original BND 100/130 bitumen according to ST RK 1225-2019 “Mixtures of asphalt concrete road, airfield and asphalt concrete. Technical conditions” must have a compressive strength at 20 °C of at least 2.2 MPa and at 50 °C of at least 1.0 MPa.

In comparison with the strength characteristics of an unmodified asphalt mixture sample at 20 °C (R*_D_* = 3.2 MPa), the compressive strength increased by 15.63–21.88% (∆R*_D_* = 0.5–0.7 MPa) in the sample using binary composition “bitumen-AG-4I”. The effect of increasing strength in binary compositions “bitumen-AS-1” was 6.25–12.50% compared to the use of the original binder (∆R*_D_* = 0.2–0.4 MPa). The maximum increase in compressive strength at 20 °C by 28.13% (∆R*_D_* = 0.9 MPa) was achieved in an asphalt concrete sample made on the basis of a ternary bitumen composition, including the combined presence of AG-4I and AS-1 (C_AG-4I_ = 1.0 g/dm^3^; C_AS-1_ = 1.0 g/dm^3^).

A comparative analysis of the physicochemical characteristics of the binder and the structural and mechanical properties of asphalt mixture samples shows that the determining factor for strength is the average size of the bee-like structures of bitumen (Table 9). In comparison, viscosity is only a detector of associative–dissociative transformations that occur when the modifiers under study are introduced into bitumen. As a result of intermolecular interactions between additives and bee-like structures of bitumen, the size of microassociates, the density of their distribution in the dispersion medium of the binder, and, accordingly, the strength of asphalt concrete change.

## 4. Conclusions

It has been established that in binary “bitumen–surfactant” systems, the introduction of both AS-1 and AMDOR-10 into bitumen is accompanied by a decrease in the absolute viscosity over the entire range of studied additive concentrations, which indicates the development of destructuring processes as a result of the inclusion of amphiphilic compounds in the dispersed structure bitumen. The lowest degree of degradation corresponds to the maximum surface activity in bitumen compositions with a concentration of modifiers of 1.0 g/dm^3^.In the binary system “bitumen-AG-4I” in the area of low concentrations (C_AG-4I_ ≤ 0.5 g/dm^3^), petroleum oils cause a restructuring of the dispersed structure of bitumen, accompanied by a decrease in specific viscosity. In the region of high concentrations (C_AG-4I_ ≥ 1.0 g/dm^3^), another component of AG-4I, polyisobutylene, forms an intermolecular polymer grid in the volume of the dispersed system, as a result of which the specific viscosity increases 37 times.When adding AS-1 to the binary system “bitumen-AG-4I”, the viscosity effect of surfactants is determined by the concentration ratio of modifiers: the higher the polymer content, the higher the viscosity effect of AS-1. Based on the generalized equation and a graphical representation of the function of several variables, it is shown that the absolute viscosity value closest to the viscosity of the origin bitumen (*η* = 61 s) belongs to a triple composition containing 1 g/dm^3^ AG-4I and 1 g/dm^3^ AS-1 (*η* = 57 s).Using the AFM method, it is established that in binary systems, the depth of dispersed changes in the microstructure of bitumen under the influence of the modifier is maximum in the presence of AG-4I. As the concentration of AG-4I increased from 0 to 0.5 g/dm^3^, the content of fine fractions (≤2.0 µm) increased by 1.7 times, and the average size of “asphaltene-resin” aggregates decreased from 3.69 to 2.23 µm.It has been established that the condition for achieving the minimum size of bee-like structures is the introduction of 1.0 g/dm^3^ AG-4I and 1.0 g/dm^3^ AS-1 into the bitumen; the average size of dispersions is 1.66 µm. In these concentration regimes, as a result of simultaneous exposure to AG-4I and AS-1, fractions with a size of more than 4.0 µm were completely destroyed, and aggregates in the range 2.0–4.0 µm were destroyed two times; the content of fine fractions (≤2.0 µm) increased by 57.4% compared to virgin bitumen and amounted to 81.9%.A close correlation was revealed in the nature of changes in the dispersed composition of modified bitumen and the strength indicators of asphalt concrete samples. In the asphalt mixture sample made on the basis of the ternary composition “bitumen-AG-4I-AS-1” (C_AG-4I_ =1.0 g/dm^3^; C_AS-1_ = 1.0 g/dm^3^), the maximum increase in compressive strength R*_D_* was achieved with the smallest size of bee-like structures of modified bitumen. This shows that the modifying role of additives is in the formation of dense, durable asphalt concrete, which is achieved due to the deep disaggregation of bitumen microdispersions and their uniform distribution over the entire volume of the binder.

## Figures and Tables

**Figure 1 polymers-16-01370-f001:**
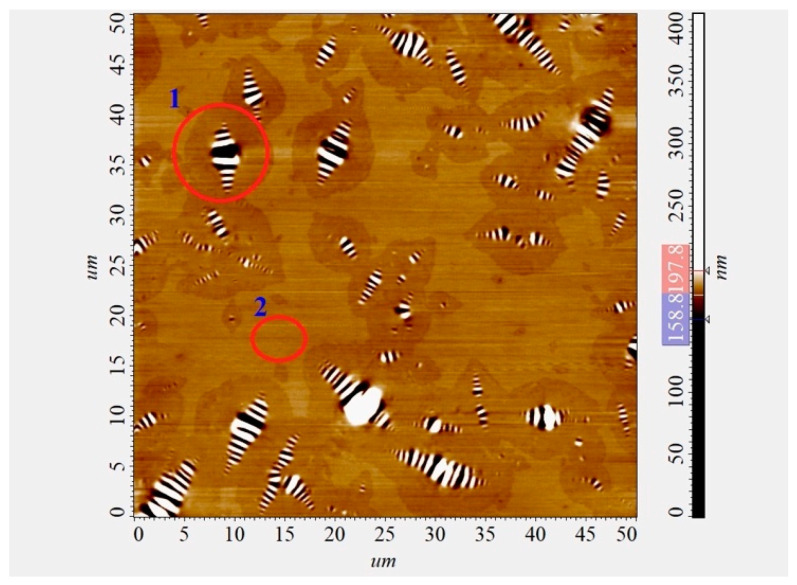
AFM images of the surface of bitumen samples. Important areas: 1—bee-like structure, 2—continuous phase.

**Figure 2 polymers-16-01370-f002:**
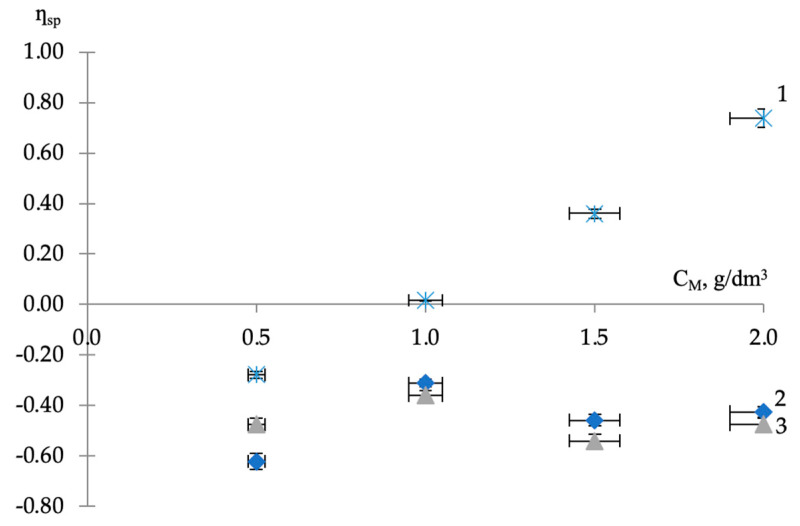
Dependence of the specific viscosity of binary compositions on the concentration of modifiers: 1—AG-4I, 2—AS-1, 3—AMDOR-10.

**Figure 3 polymers-16-01370-f003:**
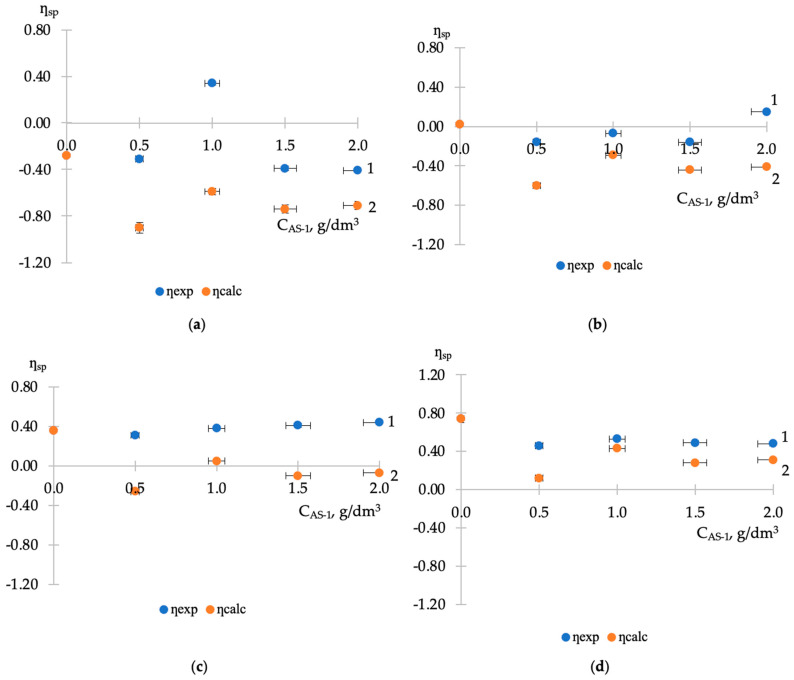
Dependence of experimental (1) and calculated (2) specific viscosity of ternary systems with a fixed content of AG-4I on the concentration of AS-1. C_AG-4I_, g/dm^3^: (**a**) 0.5; (**b**) 1.0; (**c**) 1.5; (**d**) 2.0.

**Figure 4 polymers-16-01370-f004:**
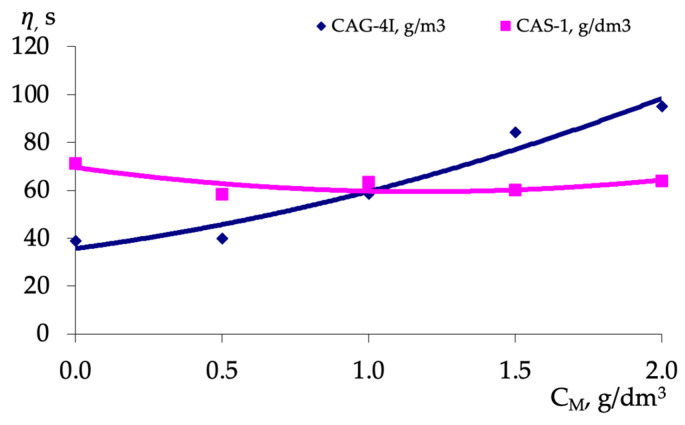
The effect of the additive concentration on the viscosity of the bitumen mixture.

**Figure 5 polymers-16-01370-f005:**
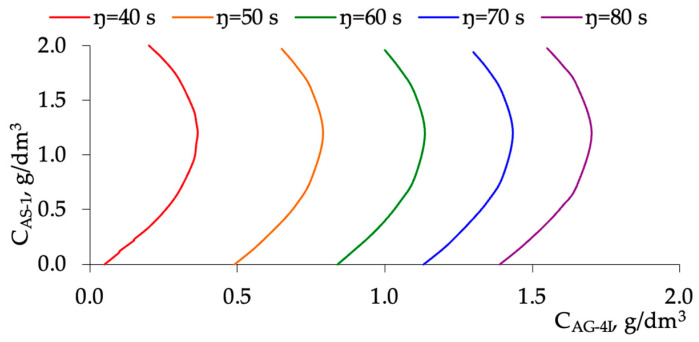
Nomograms for choosing the optimal concentration ratios of polymer AG-4I and surfactant AS-1 to achieve the required viscosity of the bitumen mixture (40, 50, 60, 70, or 80 s, respectively).

**Figure 6 polymers-16-01370-f006:**
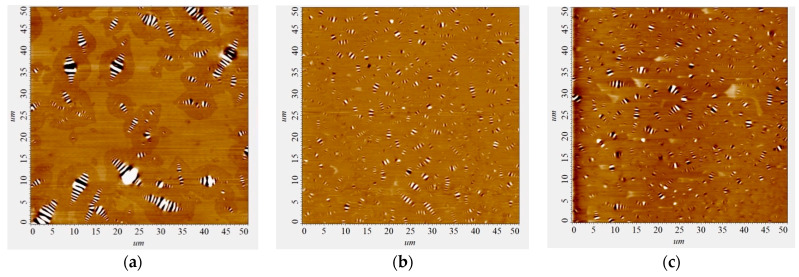
AFM images of the surface of bitumen samples: (**a**)—original bitumen; (**b**)—bitumen + 0.5 g/dm^3^ AG-4I; (**c**)—bitumen + 2.0 g/dm^3^ AG-4I.

**Figure 7 polymers-16-01370-f007:**
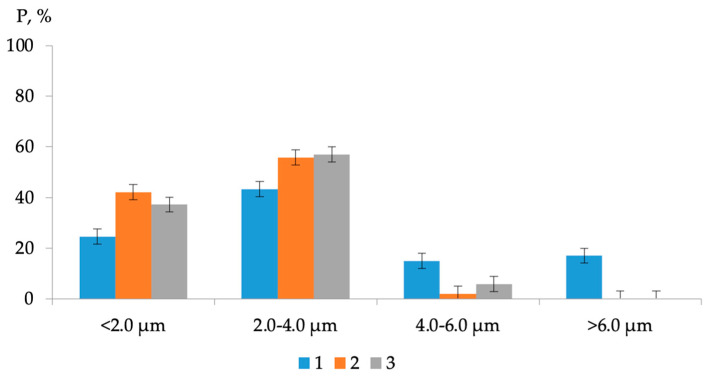
Distribution of microdispersions of bitumen binder by size class in compositions with different contents of AG-4I: 1—0 g/dm^3^; 2—0.5 g/dm^3^; 3—2.0 g/dm^3^.

**Figure 8 polymers-16-01370-f008:**
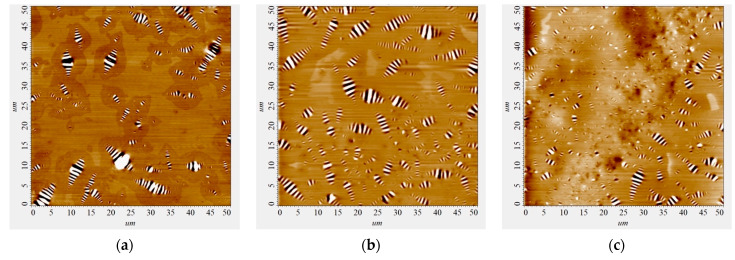
AFM images of the surface of bitumen samples: (**a**)—original bitumen; (**b**)—bitumen + 0.5 g/dm^3^ AS-1; (**c**)—bitumen + 1.0 g/dm^3^ AS-1.

**Figure 9 polymers-16-01370-f009:**
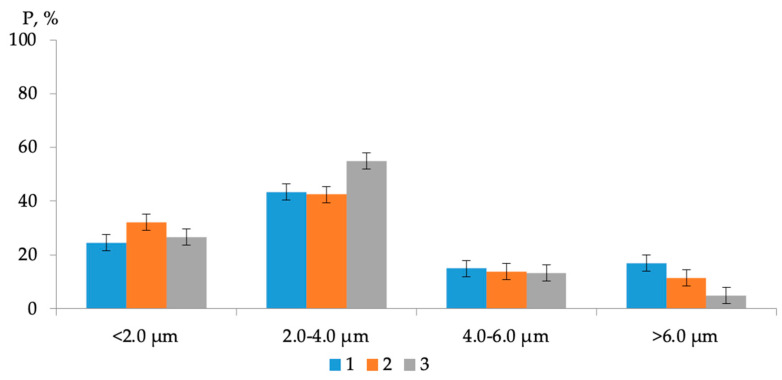
Distribution of microdispersions of bitumen binder by size class in compositions with different AS-1 contents: 1—0 g/dm^3^; 2—0.5 g/dm^3^; 3—1.0 g/dm^3^.

**Figure 10 polymers-16-01370-f010:**
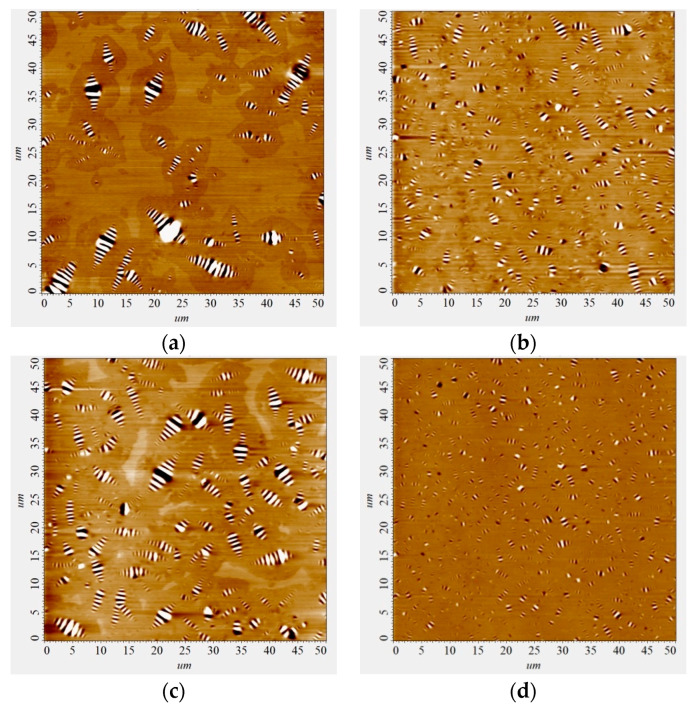
AFM images of the surface of bitumen samples: (**a**)—original bitumen; (**b**)—bitumen + 0.5 g/dm^3^ AG-4I + 1.0 g/dm^3^ AS-1; (**c**)—bitumen + 2.0 g/dm^3^ AG-4I + 1.0 g/dm^3^ AS-1; (**d**)—bitumen + 1.0 g/dm^3^ AG-4I + 1.0 g/dm^3^ AS-1.

**Figure 11 polymers-16-01370-f011:**
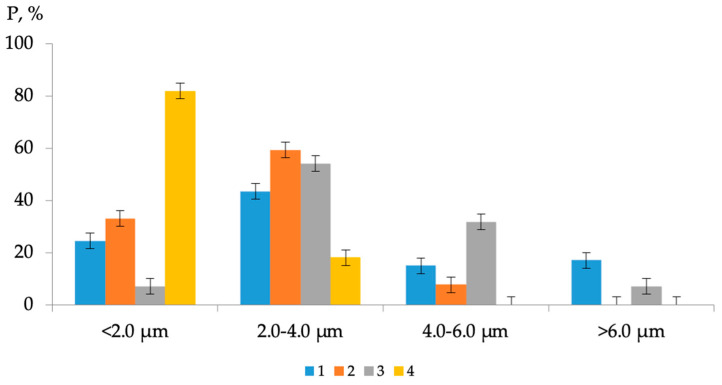
Distribution of microdispersions of bitumen binder by size class in ternary compositions: 1—original bitumen; 2—bitumen + 0.5 g/dm^3^ AG-4I + 1.0 g/dm^3^ AS-1; 3—bitumen + 2.0 g/dm^3^ AG-4I + 1.0 g/dm^3^ AS-1; 4—bitumen + 1.0 g/dm^3^ AG-4I + 1.0 g/dm^3^ AS-1.

**Table 1 polymers-16-01370-t001:** Bitumen samples for microstructural analysis.

Sample Number	Components of the Composition
1	Original bitumen
2	Bitumen + 0.5 g/dm^3^ AS-1
3	Bitumen + 1.0 g/dm^3^ AS-1
4	Bitumen + 0.5 g/dm^3^ AG-4I
5	Bitumen + 2.0 g/dm^3^ AG-4I
6	Bitumen + 0.5 g/dm^3^ AG-4I + 1.0 g/dm^3^ AS-1
7	Bitumen + 2.0 g/dm^3^ AG-4I + 1.0 g/dm^3^ AS-1
8	Bitumen + 1.0 g/dm^3^ AG-4I + 1.0 g/dm^3^ AS-1

**Table 2 polymers-16-01370-t002:** Rheological characteristics of binary bitumen compositions.

C_M_, g/dm^3^	AG-4I		AS-1		AMDOR-10	
	*η*, s	*η_rel_*	*η*, s	*η_rel_*	*η*, s	*η_rel_*
0	61	1.00	61	1.00	61	1.00
0.5	44	0.72	23	0.38	32	0.52
1.0	62	1.02	42	0.69	39	0.64
1.5	83	1.36	33	0.54	28	0.46
2.0	106	1.74	35	0.57	32	0.52

**Table 3 polymers-16-01370-t003:** Influence of quantitative contents of AG-4I on the rheological characteristics of “bitumen-AG-4I” compositions.

C_M_, g/dm^3^	*η_sp_*	Δ*η_sp_*
0.5	−0.26	+0.28
1.0	0.02	
1.0	0.02	+0.34
1.5	0.36	
1.5	0.36	+0.38
2.0	0.74	

**Table 4 polymers-16-01370-t004:** Influence of quantitative contents of modifiers on the rheological characteristics of bitumen–surfactant compositions.

C_M_, g/dm^3^	“Bitumen-AMDOR-10” System	
	*η_sp_*	Δ*η_sp_*
0.5	−0.48	+0.12
1.0	−0.36	
1.5	−0.54	+0.06
2.0	−0.48	
**C_M_, g/dm^3^**	**“Bitumen-AS-1” System**	
0.5	−0.62	+0.31
1.0	−0.31	
1.5	−0.46	+0.03
2.0	−0.43	

**Table 5 polymers-16-01370-t005:** Influence of quantitative contents of AG-4I on the rheological characteristics of ternary compositions “bitumen-AG-4I-AS-1”.

	C_AG-4I_ = 0.5 g/dm^3^		C_AG-4I_ = 1.0 g/dm^3^	
C_AS-1_, g/dm^3^	*η* * _sp_ *	Δ*η* *_sp_*	*η* * _sp_ *	Δ*ƞ* *_sp_*
0.5	−0.16	+0.50	−0.16	+0.11
1.0	0.34		−0.07	
1.0	0.34	−0.73	−0.07	−0.09
1.5	−0.39		−0.16	
1.5	−0.39	−0.02	−0.16	+0.31
2.0	−0.41		0.15	
	C_AG-4I_ = 1.5 g/dm^3^		C_AG-4I_ = 2.0 g/dm^3^	
0.5	0.31	+0.07	0.46	+0.07
1.0	0.38		0.53	
1.0	0.38	−0.07	0.53	−0.07
1.5	0.31		0.46	
1.5	0.31	+0.02	0.46	+0.02
2.0	0.44		0.48	

**Table 6 polymers-16-01370-t006:** Orthogonal plan of a two-factor experiment at 5 levels (5 × 5) to study the effect on the viscosity of a bitumen mixture of concentrations of jointly present AG-4I and AS-1.

n	C_AG-4I_, g/dm^3^	C_AS-1_, g/dm^3^	Viscosity, *ƞ*, s
1	0	0	61
2	0	0.5	23
3	0	1.0	42
4	0	1.5	33
5	0	2.0	35
6	0.5	0	44
7	0.5	0.5	42
8	0.5	1.0	40
9	0.5	1.5	37
10	0.5	2.0	36
11	1.0	0	62
12	1.0	0.5	51
13	1.0	1.0	57
14	1.0	1.5	53
15	1.0	2.0	70
16	1.5	0	83
17	1.5	0.5	80
18	1.5	1.0	84
19	1.5	1.5	86
20	1.5	2.0	88
21	2.0	0	106
22	2.0	0.5	95
23	2.0	1.0	93
24	2.0	1.5	91
25	2.0	2.0	90

**Table 7 polymers-16-01370-t007:** The effect of the concentration of the polymer AG-4I on the viscosity (ŋ, s) at a temperature of 130 °C.

n	C_AG-4I_, g/dm^3^				
	0	0.5	1.0	1.5	2.0
1	61	44	62	83	106
2	23	42	51	80	95
3	42	40	57	84	93
4	33	37	53	86	91
5	35	36	70	88	90
The average value	38.8	39.8	58.6	84.2	95.0

**Table 8 polymers-16-01370-t008:** The effect of the concentration of surfactant AS-1 on the viscosity (ŋ, s) at a temperature of 130 °C.

n	C_AS-1_, g/dm^3^				
	0	0.5	1.0	1.5	2.0
1	61	23	42	33	35
2	44	42	40	37	36
3	62	51	57	53	70
4	83	80	84	86	88
5	106	95	93	91	90
The average value	71.2	58.2	63.2	60.0	63.8

**Table 9 polymers-16-01370-t009:** Compressive strength and microstructural characteristics of modified bitumen compositions.

Modifier	C_M_, g/dm^3^	Bitumen			Asphalt Mixture	
		*ƞ*, s	*a_av_*	P ≤ 2 µm, %	R*_D_* (20 °C), MPa	R*_D_* (50 °C), MPa
	0	61.0	3.69	24.5	3.2	1.1
AG-4I	0.5	44.0	2.23	42.2	3.9	1.4
	1.0	62.0	2.38	37.2	3.7	1.3
AS-1	0.5	23.0	3.14	32.2	3.4	1.1
	1.0	42.0	3.04	26.7	3.6	1.4
AG-4I + AS-1	0.5 + 1.0	40.0	2.78	33.0	3.7	1.3
AG-4I + AS-1	1.0 + 1.0	57.0	1.66	81.9	4.1	1.8
AG-4I + AS-1	2.0 + 1.0	93.0	3.83	7.1	3.0	1.0

## Data Availability

The original contributions presented in the study are included in the article, further inquiries can be directed to the corresponding authors.

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
