# Peer review of "The Effect of the Microstructure and Viscosity of Modified Bitumen on the Strength of Asphalt Concrete"

_polymers, 2024, doi:10.3390/polym16101370_

Round 1

Reviewer 1 Report

Comments and Suggestions for Authors

This manuscript is interesting but requires significant corrections. Moreover, it must be formatted correctly because it is very difficult to read. The descriptions in the methodology and the discussion of the results also require corrections. This all adds up to quite significant corrections.

Detailed comments below:

Line 102: In my opinion, this justification for conducting research is insufficient. There should also be an additional scientific thread here. Extend this part of your work.

Line 106: These bullet points of individual work stages are unnecessary. Here it is enough to provide only a short scope of work.

Line 165: Such descriptions justifying the need to perform research should not be written in the methodology, especially since these are commonly performed tests.

Line 171: This is correct.

Line 187: Failure to provide the name of the testing machine and its parameters during testing. This needs to be completed.

Line 193: Did you use any software here? If so, please enter it.

Line 207: Discussion of results requires completion. There are no comparisons of the obtained results with studies by other authors. Whether the results obtained are close or far from the expected values. what is the typical strength of the bituminous mixtures used. What parameters influence such strength, etc.

Figure 1 - 11: x, y axis descriptions are difficult to read or uneven. Adapt to the magazine's requirements.

Figure 1: In my opinion, photos of the equipment are unnecessary.

Figure 10: mark the areas in the drawing that are important (from the point of view of your research).

Figure 11: Add error whiskers to the graph bars.

Line 554: The research conclusions are too precise. These are basically descriptions of the results obtained. You should formulate them correctly. Of course, a significant part can remain, but it needs to be improved.

References: Complete the necessary literature.

Author Response

Thank you for your comments. Responses are in attached file.

Reviewer 2 Report

Comments and Suggestions for Authors

Dear Authors, according to the stated purpose, this article written about the research  to establish the influence of the microstructural and rheological characteristics of modified bitumen compositions on the strength indicators of asphalt concrete. There are suggestions how to improve this article, provided below:

1. Introduction must be with details about the novelty of this research.

2.  Methodology can be extended with specific ingredients for the composition of asphalt concrete.

3. Results can be rearranged in investigated characteristics and chronologic homologically obtained results order.

4. Conclusions are written with results section statements, without any recommendation according to future possibilities to use the obtained results.

5. References dated 1996 can be changed to newest obtained from global research journals articles for this topic.

Sincerely, Reviewer.

Author Response

Thank you for your comments! Responses are in attached file. 

Round 2

Reviewer 1 Report

Comments and Suggestions for Authors

The authors corrected most of the comments included in the review. After minor corrections, the article can be published.